# Modelling of Gyratory Crusher Liner Wear Using a Digital Wireless Sensor

**DOI:** 10.3390/s23218818

**Published:** 2023-10-30

**Authors:** Tao Ou, Wei Chen

**Affiliations:** 1School of Mechanical and Vehicle Engineering, Hunan University, Changsha 410012, China; outao@hnu.edu.cn; 2School of Intelligent Manufacturing Ecosystem, Xi’an Jiaotong-Liverpool University, Suzhou 215123, China; 3School of Energy Science and Power Engineering, Central South University, Changsha 410017, China

**Keywords:** wear measurement, intrusive sensor, discrete element modelling, gyratory crusher

## Abstract

A gyratory crusher is a key mineral processing asset in a comminution circuit. Monitoring and predicting the crusher liner wear is essential to ensure the throughput and product quality are maintained during production. This study developed a digital sensor and a discrete element modelling (DEM)-coupled methodology to monitor and reconstruct the gyratory crusher concave liner wear pattern. The developed digital sensor was able to track and report the live thickness of the specific installation point on a concave liner during operation. A wear reconstruction model was then developed based on the wear intensity obtained using the DEM and digital sensor results. The wear reconstruction model predictive results were subsequently compared with site measurements after 95 days of operation. The results indicated that the wear reconstruction model showed good agreement with measured results in terms of wear zone distribution as well as quantitative wear rate prediction. The outcome of this study can be potentially utilised in the mineral processing industry for plant monitoring and automation.

## 1. Introduction

A subject of particular importance to the resources industry concerns the effective comminution operation for run-of-mine (ROM) ore materials. A gyratory crusher is often utilised at the very beginning of the comminution circuit [1,2,3] to break large rocks into a 120~150 mm size range, which prepares the ROM ores for semi-autogenous (SAG) or autogenous (AG) milling [4]. As shown in Figure 1, the size of the rock fed into the gyratory crusher is continuously reduced from the feed end to the discharge end within a crushing chamber.

The product throughput is critical when selecting and determining the performance of a gyratory crusher. Under a fixed mechanical drive system and feed size, the product throughput is predominantly determined by the specific energy (kWh/t) of the rock [5,6], the mantle rotational speed and the closed side setting (CSS). Historically, the crushability test [7,8,9] was commonly performed to determine the specific energy level of an ore sample. However, this test was often conducted on the drill core samples where the representation of the actual rock shape and size are limited, and this results in an inaccurate estimation of the rock’s specific energy. Subsequently, the performance of the gyratory crusher is then affected.

As a critical safety and reliability performance metric, a number of methods have been developed to track the crusher liner wear. Manual ultrasonic testing has been widely utilised in practice to report wear conditions. However, this requires human entry into the crusher chamber and poses significant safety risks as well as production loss due to the shutdown of the crusher. Manual testing has recently been replaced with the spatial laser scanning method, with which the surface geometry of the worn liners is captured with point cloud data collected using a scanner. The wear is then estimated by comparing the worn liner geometry with the original as-installed condition [10]. Nevertheless, this also requires production shutdown and uninstallation of the central mantle. Liner wear has also been estimated based on indirect metrics, such as monitoring the product size distribution and the power draw of the mechanical drive [11,12,13,14]. However, these metrics are also impacted by the ore properties and operating strategy, which only provide qualitative measures of liner wear. To the knowledge of the authors, there has been no solution reported for online wear monitoring of gyratory crusher liners.

Recent advancements in discrete element modelling (DEM) have enabled the simulation of industrial-scale gyratory crushers [15,16,17]. Particle breakage models were developed and incorporated into DEM to study the size reduction process of rock. Bruchmuller [18] developed a sphere breakage model based on the critical energy level, breakage probability and a progeny size parameter, and its application in a milling process showed promising results. Cleary [19,20,21] utilised a superquadrics-based particle shape to model the non-spherical particle breakage in crushing and milling applications. Furthermore, the bonded-particle method [22] was also developed to utilise sphere clumps to model progressive size reduction in non-spherical particles. Nevertheless, numerical modelling typically accounts for no effect of the liner wear, which requires improvement before accurate prediction is achieved.

The actual wear performance of the concave liner is determined by mixed effects of mill operational metrics, ore properties and the liner alloy selection. Increasing the mantle power and speed typically results in a higher wear rate due to an increase in rock–liner interactions inducing elevated abrasion and impact wear stresses [11,12,23,24]. More importantly, the physical, geomechanical and metallurgical properties of rock also significantly affect liner wear. Highly angular particles generally cause higher wear, as well as higher quartz concentration in the ore deposits, both of which cannot be maintained due to upstream orebody variations. Additionally, the liner’s wear life also largely relies on its material selection. Materials with higher abrasion wear resistance, such as high manganese alloy [25], are often selected for such applications due to high impact stress. Therefore, accurate modelling and prediction of concave liner wear remains a challenge in the mineral processing industry.

This study aims to develop a coupled digital sensing and numerical modelling technique to provide accurate prediction and monitoring solutions for gyratory crusher liners while in operation. A live concave liner thickness sensing technique is developed and embedded into the metal casting (proposed installation positions shown in Figure 1b). A discrete element modelling incorporated with a breakage model is developed for modelling the crushing process as well as global wear distribution. A wear reconstruction algorithm is developed by combining the sensor measurements and the wear distribution from numerical modelling.

## 2. Wear Sensor Development

To measure the live concave liner thickness, an intrusive digital sensor was developed and embedded into the concave castings. As shown in Figure 2a, a series of loops was configured onto a narrow circuit board. Each circuit loop represents a specific distance/thickness value. During operation, when a circuit loop is damaged, a corresponding distance/thickness value is then reported. The distance between two circuit loops was controlled at 1 mm, which is the resolution of the measurement. Because all circuit loops were controlled in a cylindrical space with a diameter of 5 mm (Figure 2b), its cross-sectional area was considered infinitely small compared with the wear liner surface [26]. Thus, the sensor was presumed to wear at the same rate as the wear material surrounding it. And because the accuracy of the sensor was controlled by the manufacturing tolerance of the circuit loops, the sensor also exhibited the advantage of not requiring calibration.

The status of each circuit loop is monitored with on-board units and a central microcontroller unit (MCU) [27]. The sensor was also powered with an external battery unit. The status of all loops and a corresponding thickness value were then transmitted to an external Zigbee [28,29] wireless transmitter via a serial peripheral interface (SPI) [30]. The wireless transmitter then forwarded the data to a central cloud app via a 4G gateway for reporting purposes.

The sensor board was typically encapsulated with polyurethane material. As shown in Figure 2c, in a typical installation, the measurement bit could be inserted into the metal casting, with its power and data cable routed out of the main mechanical structure for data transmission.

In terms of the site trial, a METSO Nordberg 62–75 gyratory crusher was selected for this study. To install the developed sensor into the concave metal casting, a sensor insert was manufactured to accommodate the wear sensor above, as shown in Figure 3a. A sensor fastener was designed to lock the sensor inserted into the concave liners after installation. Ten sensors were installed on the high wear zone of the concave liners, as shown in Figure 3b. All sensors were distributed from the top row to the bottom row of the concave liners, as shown in Figure 3c. The averaged sensor readings for each row of installation were reported.

## 3. Numerical Modelling Programme

The aforementioned wear sensor is designed to measure the live thickness in high wear zones of a concave liner. It is practically impossible and structurally hazardous to incorporate a large number of wear sensors. Thus, a DEM-based numerical method was utilised to model the global wear distribution on the entire concave liner, with which the wear sensor measurements can then be coupled to quantitatively reconstruct the global wear pattern.

### 3.1. Discrete Element Modelling

An in-house DEM code was utilised for modelling the gyratory crusher in this study. The Hertz–Mindlin model [31,32] is often used to compute the particle–particle and particle–wall contacts. The contact force between two particles includes a normal force ($F_{n}$) component and a tangential force ($F_{t}$) component
(1)$$\begin{array}{c} F=F_{n}+F_{t}\\ F=\left(k_{n}{\delta n}_{ij}-\gamma_{n}{vn}_{ij}\right)+\left(k_{t}{\delta t}_{ij}-\gamma_{t}{vt}_{ij}\right) \end{array}$$
where

$F_{n}$ is the normal contact force;$F_{t}$ is the tangential contact force;$k_{n}$ is the elastic stiffness for normal contact;${\delta n}_{ij}$ is the normal overlap;$\gamma_{n}$ is the viscoelastic damping constant for normal contact;${vn}_{ij}$ is the normal relative velocity (normal component of the relative velocity of the two particles);$k_{t}$ is the elastic stiffness for tangential contact;${\delta t}_{ij}$ is the tangential overlap;$\gamma_{t}$ is the viscoelastic damping constant for tangential contact;${vt}_{ij}$ is the tangential relative velocity (tangential component of the relative velocity of the two particles).

For detailed formulations of the DEM principle, readers are directed to past studies [31]. In this study, a rolling friction model was also added to the DEM framework to account for the particle shape effect of the rocks, and the elastic–plastic spring–dashpot (EPSD) model was utilised [33,34]. This model adds an additional torque contribution in an incremental way as follows
(2)$$M_{r,t+∆t}^{k}=\left\{\begin{array}{cc} M_{r,t}^{k}-k_{r}\omega_{r}∆t; & \mathrm{if}\;\left|M_{r,t}^{k}-k_{r}\omega_{r}∆t\right|<\mu_{r}R^{*}\left|F_{n}\right|\\ \mu_{r}R^{*}\left|F_{n}\right|\frac{M_{r,t}^{k}-k_{r}\omega_{r}∆t}{\left|M_{r,t}^{k}-k_{r}\omega_{r}∆t\right|}; & \mathrm{otherwise} \end{array}\right.$$
where

$k_{r}=k_{t}\;{R^{*}}^{2}$ is the rolling stiffness;$\omega_{r}$ is the relative angular velocity of the two particles in contact.

The EPSD rolling friction model is an established method to model the rolling resistance of granular rock particles. Furthermore, the Finnie wear model [35] was also selected in this study to compute the wear intensity exhibited by the material flow on the concave liners.

### 3.2. Particle Breakage Modelling

The Bruchmueller breakage model [18] was used in this study to model the fragmentation behaviour of rocks in gyratory crushers. During DEM computation, it kept track of a fragmentation criterion for each particle. Particles where fragmentation events were detected were replaced with a set of fragments. The impact energy was tracked for each particle–particle and particle–wall contact as
(3)$$E_{i(t+∆t)}=E_{i(t)}+F_{n}{vn}_{ij}dt$$
where $E_{i(t)}$ and $E_{i(t+∆t)}$ are the impact energy in the time steps t and t+∆t. The summation of the impact energy started with the contact and finished when the sign of relatively normal velocity reversed. When the computation of the impact energy was completed, it was compared with the minimum impact energy that causes damage, $E_{0}$, which was defined by the user as a material property parameter. Therefore, once
(4)$$E_{i}>E_{0}$$

Then, the surplus energy $E_{i}-E_{0}$ was added to the accumulated damage energy $E_{dmg}$. The probability for particle fragmentation was then defined as
(5)$$p=\left[1-exp(-bkp\cdot d\cdot E_{dmg})\right]$$
where bkp is the breakage probability parameter and also a material parameter. bkp was difficult to obtain for a material. However, it was suggested that the *b* parameter obtained from the JK drop weight test [36,37,38] can be utilised to approximate bkp·d. Thus, it was easy to obtain from experiments.

When fragmentation occurred, the following particle size distribution definition was utilised
(6)$$T_{10}=A\left[1-exp(-bkp\cdot d\cdot E_{dmg})\right]$$
(7)$$T_{25}=0.5T_{10}$$
(8)$$T_{50}=0.26T_{10}$$
where A is the maximum achievable $T_{10}$ [39] in a single breakage event and ranges from zero (breakage into a few fragments) to 50 (attrition-like fragmentation). $T_{10}$, $T_{25}$ and $T_{50}$ are the mass percentage of fragment particles larger than 10%, 25% and 50% of the original particle.

Utilising Equations (3)–(8), the breakage behaviour as well as the progeny size distribution after breakage can be fulfilled. For detailed modelling principles, readers are directed to the original work by Bruchmueller.

### 3.3. Numerical Setup

Based on the aforementioned DEM modelling principle, the following numerical simulation setup was proposed. As shown in Figure 4a, at the beginning of the modelling, fresh feed ore with a nominal tonnage of 3500 tph was delivered into the gyratory crusher using a belt feeder travelling at 1.5 m/s. The feed ore featured a typical copper porphyry deposit with a particle density of 2600 kg/m^3^. The feeding particle size distribution is shown in Figure 4b, which was subsequently used in the modelling setup. Once the feed ore entered the crushing chamber, a rotational–oscillatory motion was induced to the mantle to crush the rocks. The rotational–oscillatory motion was configured with a designed rotational speed of 160 rpm and a CSS of 125 mm. Once the rocks were sufficiently small compared with CSS, the product stream was then discharged out of the bottom of the crusher.

In this study, the feed and product size distributions were measured using a combination of sieving analysis and image processing. Essentially, original samples collected from the site were initially divided into −100 mm and +100 mm groups. With the −100 mm sample, standard sieving analysis [40] was performed to obtain its size distribution. The minimum sieve opening size was controlled at 16 mm, which was also selected as the cut-off size for DEM modelling due to computational restrictions when even finer particles were considered. Furthermore, for the +100 mm sample, an image processing analysis was performed to estimate its equivalent particle size. Readers are directed to the study [41] for the detailed procedure. The final particle distribution was established by combining the sieve analysis and the image processing results.

A total of 60 seconds of simulation was conducted, and the modelling parameters for DEM and the breakage model are shown in Table 1. During the modelling process, the product size distribution as well as the mass flow rate at the discharge position of the crusher were constantly tracked and compared with the size performance.

## 4. Results and Discussion

### 4.1. Digital Wear Sensor Measurements

Depending on the ore hardness, concave liner material types and the operating strategy, the actual service life of the concave liner spans from a couple of months to beyond a year. In this study, selected concave liners exhibited a typical service life of approximately 100 days. In the sensor trial campaign, the set of concave liners lasted 95 days before relining was performed.

During 95 days of the gyratory crusher operation, sensor readings were continuously collected and reported to the central cloud app. Figure 5a shows the averaged sensor readings for the LUP, LMID and LDWN sensor groups. LUP represents the sensor installation positions on the upper row, LMID represents the middle row and LDWN represents the bottom row of the concave liners. It was found that the LDWN row of the concave liners exhibited the highest wear, followed by the LMID and LUP liners. The wear on the LDWN row of concave liners showed approximately 6 times the wear on the LUP row of concave liners and double the wear on the LMID concave liners, as shown in Figure 5b for averaged wear rate comparisons. Such a performance was typically observed during normal gyratory crusher operations because the feed rocks were continuously fragmented from the entry to the discharge position, which resulted in a finer product inducing an increased abrasion wear mechanism to the concave liners. Whereas only larger rocks will induce impact wear on the top row of the concave liners, smaller rocks simply flow directly to the bottom row of concave liners before interacting with the mantle.

Furthermore, the wear on the LMID and LDWN groups of wear sensors appeared to be non-linear. An accelerated wear trend was observed for both groups of sensors, with the LDWN group being more evident. This may be due to the concave liner material characteristics, with which lower hardness castings are often produced in the internal section of the casting during the heat treatment process [43,44,45].

### 4.2. Gyratory Crusher Modelling Results

Based on the DEM modelling principle and the numerical setup scheme previously discussed, a simulation of the selected gyratory crusher was conducted. During modelling, the product size distribution of the crusher was measured and compared with the site measurement results. As shown in Figure 6a, the feed material continuously flowed into the crushing chamber and interacted with the mantle and concave liners. It was indicated that the particle size of the rock was continuously reduced from the top of the crusher towards the discharge. The top size of the discharged product stream was observed to be controlled at 125 mm, indicating the effectiveness of the breakage model implemented.

In addition, the product size distribution obtained from the numerical modelling was compared with the size measurements conducted at the site. As shown in Figure 6b, a close agreement between the simulation results and the site measurements was observed, indicating the developed numerical modelling framework, as well as the modelling parameter selections, were appropriate. Nevertheless, the mantle speed was fixed in the numerical modelling, which may not fully reflect the site operating conditions. Particularly, the mantle speed may be significantly slowed down when hard rocks are fed into the crusher. Readers are directed to studies on mantle power performance for more competent ore-feeding operating conditions [46].

It is important to note that actual interactions between the rock, concave and mantle are rather complex, and a holistic evaluation of major crusher operating parameters is required to ensure the accurate representation of the crushing mechanism. Nevertheless, because the mantle setting in this study exhibited a fixed rotational–oscillatory motion, the power draw results would not fully represent the dynamic interaction between the rock and mantle. For enhanced modelling of the crusher power performance, coupling of the DEM with the multi-body dynamics may be required to more accurately reflect the rock–mantle contacts. Readers are directed to a more in-depth study on the advanced modelling technique for gyratory crushers [47]. Secondly, the actual mass flow rate of the selected gyratory crusher varied in a large range during practical operations, subjected to fluctuating ore feed quality and operating constraints in the crushing circuit. A single, discrete simulated crushing behaviour in the modelling may not represent the actual performance of the selected gyratory crusher.

In actual operations, despite the fact that the gyratory crusher may process rocks with a broad hardness range, the resulting final wear distribution may be similar from campaign to campaign if the studied time scale were extended to the full service life. Numerical modelling only provides limited snapshots of the actual operating conditions in the full service life, whereas the wear sensor developed in this study captured the wear induced by the varied rock–liner contacts over the time scale, which obviously was limited to only the several points that could be measured. Therefore, the goal of this study was to obtain the qualitative wear pattern distribution on the concave liners, from which accurate sensor readings could then be leveraged for full quantitative prediction. The DEM modelling of the crusher was only required to produce the qualitative wear pattern due to the crushing mechanism. Hence, only the size distributions from the site measurements and simulation were compared.

During the numerical simulation, the wear profiles on the concave liners and the mantle were also obtained from the modelling [35,48]. As shown in Figure 7, it was indicated that the high wear zone on the concave liners was located at the mid and bottom rows. And the wear continued to reduce toward the top section. In comparison, the high wear zone of the mantle coincided with the concave liner wear results. Nevertheless, the mantle wear was indicated to be more evenly distributed compared with the concave liners.

### 4.3. Concave Global Wear Reconstruction

The intrusive wear sensor is able to provide point-based wear measurements for the concave liners. However, it is practically impossible as well as structurally hazardous to install more sensors on the concave liners. In order to capture and predict the global wear pattern on a concave liner, it is proposed in this study to couple the wear sensor measurements with the wear intensity results from the numerical modelling. Essentially, the wear for a specific concave liner row was scaled based on the available sensor measurements and the corresponding wear intensity results. That is,
(9)$$W_{x,y,z}=W_{i(x,y,z)}\frac{W_{s0}}{W_{i0}}$$
where $W_{s0}$ is the wear sensor measurements in mm; $W_{i0}$ is the corresponding wear intensity results obtained from numerical modelling; and $W_{i(x,y,z)}$ is the wear intensity results at a spatial position where wear sensors were not installed.

Using Equation (9), the global wear distribution shown in Figure 7a can then be reconstructed and converted to a quantitative wear pattern, as shown in Figure 8b. The end-of-life wear pattern on the concave wear liner was also obtained using a spatial laser scanner, and the results are shown in Figure 8a. From the comparison, it was indicated that the reconstructed wear pattern exhibited general agreement with the laser scanner results, suggesting the validity of the proposed method.

Further validation of the proposed method was also conducted by sampling the reconstructed wear results in line B-B’ with similarly positioned wear results from the laser scanner in line A-A’. As shown in Figure 9, the reconstructed wear results showed good agreement with the laser scanner results across the top to the bottom of the concave liners with a 4.7 mm maximum difference. Relatively larger discrepancies were observed between the measured results and the predicted results from the middle of the concave liner towards the concave bottom, in which predicted wear was higher than the measured results. This may be due to the distinct wear performance of the manganese liner material as well as the variation in feed. It is important to note that the manganese material would form a hardened martensitic layer upon impact, which enhances the abrasion wear resistance of the material. The middle row of the concave liners typically receives a large impact from the particle due to the crushing mechanism. However, the wear intensity modelling method in DEM accounts for no effect of martensitic layer transformation in modelling.

Consequently, it is suggested that the proposed method can be utilised for online wear monitoring and prediction during practical gyratory crusher operations. Nevertheless, it is also important to note that the wear reconstruction method proposed and validated above assumed that wear on the concave liner progressively propagated to the final worn conditions. However, in actual operation, there might be instances where a large portion of the material is suddenly removed from the liner surface due to oversized feed rock or heterogeneous liner material quality, which the proposed method would not be able to capture and predict.

## 5. Conclusions

A comprehensive investigation was carried out to develop a digital wear sensor and discrete element modelling-coupled method to monitor and predict crusher liner wear patterns. A digital sensor was developed to monitor and report the live thickness of the concave liner in a gyratory crusher. DEM modelling of the gyratory crusher was also conducted, from which the digital sensor reading was utilised to quantitatively reconstruct the concave wear profile. This study yielded the following major findings:The concave liner wear exhibited a non-linear correlation, which was initially slow and gradually ramped up towards the end of liner life.Concave liners near the discharge position of the gyratory crusher showed higher wear. And the wear continued to reduce towards the feeding position.The highest wear position in the mantle was indicated to occur at a similar position compared with concave liners; however, the wear on the mantle was more evenly distributed.Wear evolution results obtained using coupling digital sensor results and DEM wear modelling showed good agreement with laser scan measurement.

Consequently, the method developed in this study can be potentially adopted as a concave liner wear live monitoring solution and plant automation.

## Figures and Tables

**Figure 1 sensors-23-08818-f001:**
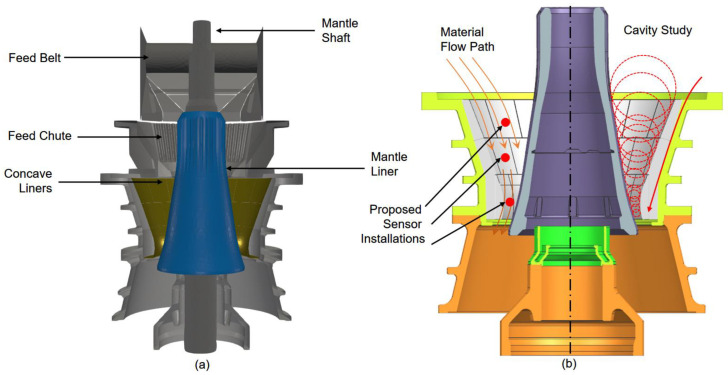
(**a**) Schematic of a gyratory crusher and its associated components. (**b**) Material flow path and principle of particle size reduction along the flow path.

**Figure 2 sensors-23-08818-f002:**
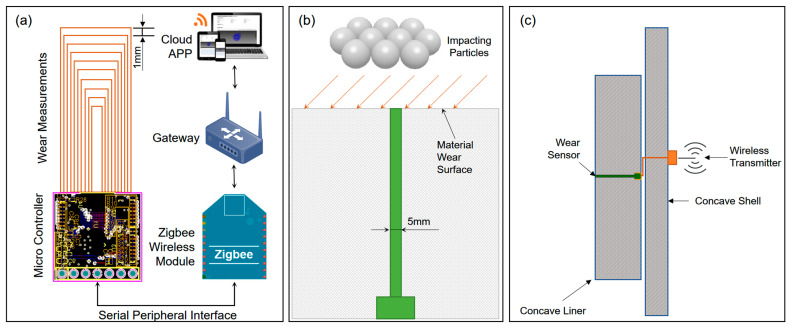
Schematic of the wear sensor design and monitoring system. (**a**) Sensor measurement principle and associated data transmitting components. (**b**) Integration of the wear sensor into the wear materials and its contacts with the wear medium (particles). (**c**) Wear sensor installation method and cable routing in the concave structure.

**Figure 3 sensors-23-08818-f003:**
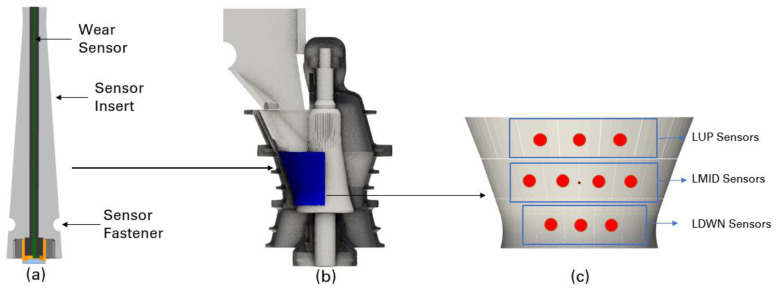
Schematic of the sensor installation and distribution on concave liners. (**a**) Sensor insert design and mechanical installation. (**b**) Selected concave liner for sensor installations. (**c**) Sensor installation positions on concave liners.

**Figure 4 sensors-23-08818-f004:**
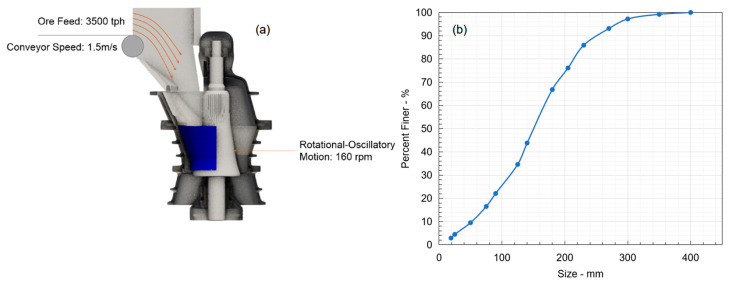
(**a**) Schematic of the DEM modelling setup. (**b**) Feed ore particle size distribution.

**Figure 5 sensors-23-08818-f005:**
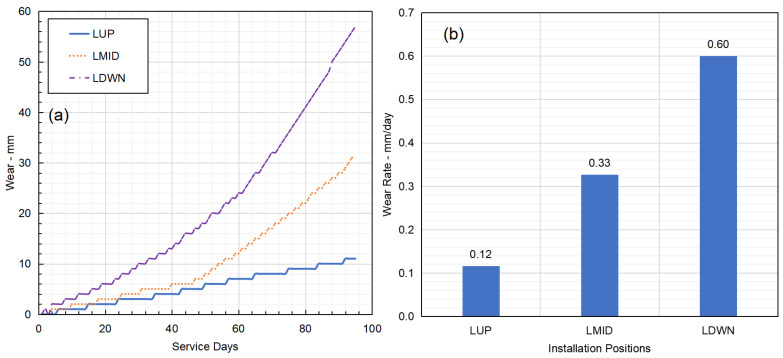
(**a**) Time series wear sensor readings from the LUP, LMID and LDWN row of concave liners. (**b**) Averaged wear rate in terms of mm/day for the three sensor groups.

**Figure 6 sensors-23-08818-f006:**
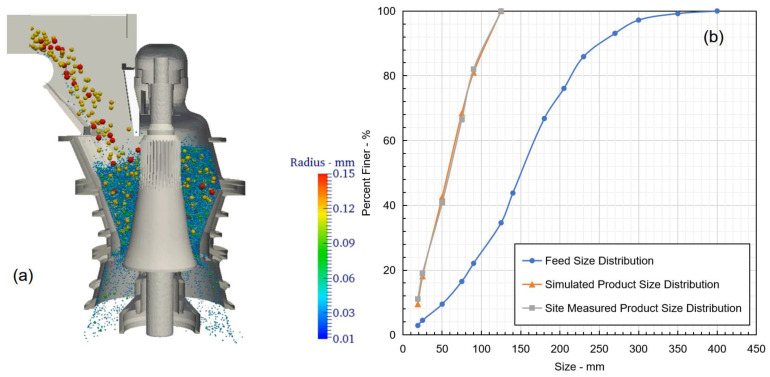
DEM modelling results of the gyratory crusher. (**a**) Particle size reduction and the flow pattern from feed to discharge. (**b**) Product particle size distribution comparison between feed, simulated and site-measured results.

**Figure 7 sensors-23-08818-f007:**
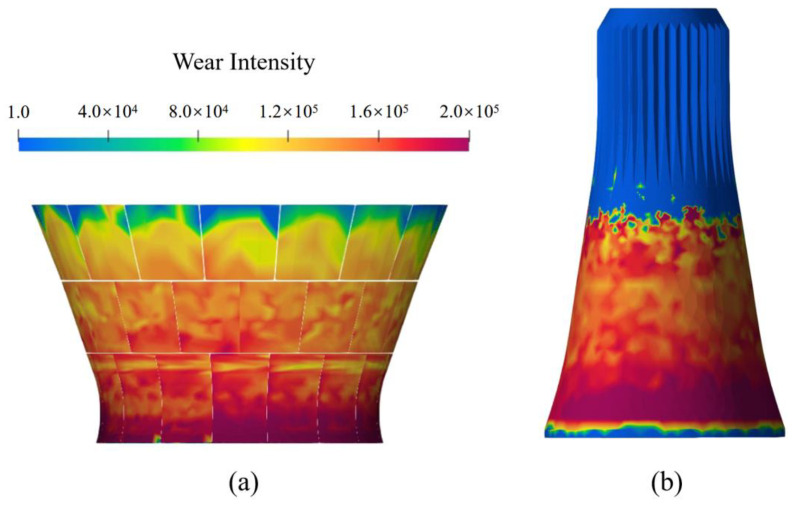
DEM wear modelling results of the gyratory crusher. (**a**) Wear intensity results on the concave liners. (**b**) Wear intensity results on the mantle surface.

**Figure 8 sensors-23-08818-f008:**
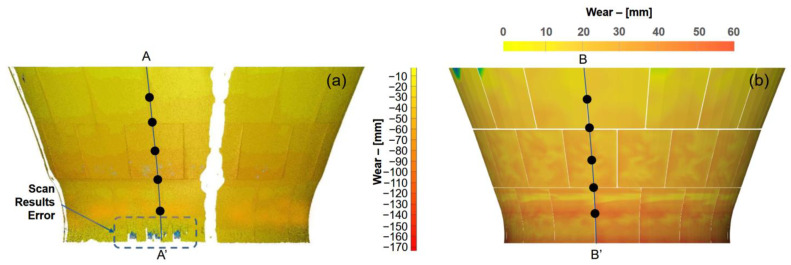
(**a**) Wear pattern on the concave liner after change out obtained using a laser scanner. (**b**) Reconstructed wear pattern results using the method proposed in this study.

**Figure 9 sensors-23-08818-f009:**
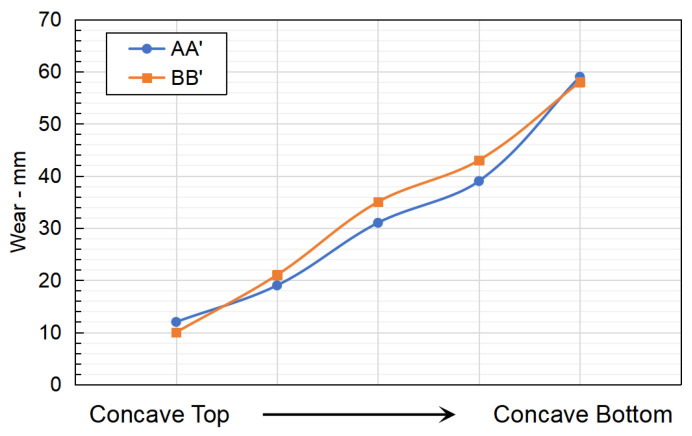
Quantitative comparison between scan wear results and wear reconstruction results in selected locations.

**Table 1 sensors-23-08818-t001:** Modelling parameters used in the numerical modelling programme.

Variable	Value	Units
Particle density	2600	kg/m^3^
Interparticle friction coefficient [42]	0.5	-
Wall friction coefficient [42]	0.3	-
Rolling friction coefficient [42]	0.3	-
Restitution coefficient	0.3	-
Poisson’s ratio	0.3	-
Young’s modulus	1 × 10^7^	Pa
$\mathrm{Minimum}\;\mathrm{impact}\;\mathrm{energy}—E_{0}$ [18]	3	J/kg
Breakage probability parameter—bkp [18]	0.9	kg/Jm
$\mathrm{Maximum}\;\mathrm{achievable}\;T_{10}$—A [18]	10	-
Time step	1 × 10^−6^	s

## Data Availability

The data are available in a publicly accessible repository. The data presented in this study are openly available in Crusher Wear Sensor Data. figshare. Dataset. https://figshare.com/articles/dataset/Crusher_Wear_Sensor_Data/24210996 (accessed on 15 September 2023).

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
