# Peer review of "Modelling of Gyratory Crusher Liner Wear Using a Digital Wireless Sensor"

_sensors, 2023, doi:10.3390/s23218818_

Round 1
Reviewer 1 Report
Comments and Suggestions for Authors
1. It appears that more detailed description on the wear sensor is required if the title is to be unchanged. Including
1) the horizontal wire gap value in Figure 2(a): for wear sensing resolution
2) the wear sensor installation point in Figure 1.
2. Most part of the paper is about the wear modeling. A better title for the paper would be 'Modeling of Gyrator Crusher Liner Wear Using A Digital Wireless Sensor'.
3. Review of existing works is necessary.
- On the gyrator crusher liner wear modeling
- On the digital wear sensor.
If no previous work exist, it needs to be said.
Reviewer 2 Report
Comments and Suggestions for Authors
Your work is interesting and shows the great performance. Please check the grammars more.
Comments on the Quality of English LanguageBased on the quality of this article, I am glad to recommend it to Sensors.
Author Response
Authors are grateful for the comments made by the reviewer. The manuscript has been further improved for grammatical errors.
Reviewer 3 Report
Comments and Suggestions for Authors
This paper presents a model that aligns well with measurements, both in terms of wear zone distribution and quantitative wear rate prediction. This is a positive indication of the model's reliability and its potential to enhance the management of gyratory crusher performance.
1. Is 95 days of operation enough? Is there any standard way and time to evaluate the reliability?
2. Does this sensor system need calibration when used in different mineral processing plants?
3. The accuracy of the wear reconstruction model depends on the accuracy of the DEM. Any limitations or assumptions in the DEM could potentially affect the reliability of the wear predictions. Please comment on this possible weak point.
4. Please add more discussion on the discrepancy between the simulated and measured results.
5. How does this work advance the existing works that seem missing.
Reviewer 4 Report
Comments and Suggestions for Authors
Major:
- There is no experimental evidence of the feed size distribution and product size distribution. In the methodology, it is not clear how this information was measured and obtained. Then, in Figure 6b, a comparison between the experimental and simulated product size distribution is presented, having a good match between both. The minimum simulated particle size is not presented (which limits the simulated PSD) and neither the series of sieves used to obtain the experimental PSD. Besides, it seems that the feed particle size distribution is too fine for the size of the crusher. I ask the authors to clarify these points.
- The validation of the model is incomplete, just the PSD is presented. The physics inside a crusher chamber is much more complex and involves variables such as power and mass flow that must be compared for validation.
Minor:
Line 36: Please check the sentence because the product throughput also depends on the rotational speed and closed side setting.
Figure 1: Please add the corresponding references of the figures.
Line 61: Please add the missing reference before index.
Line 61 to 68: Please add the missing reference:
https://www.sciencedirect.com/science/article/pii/S0032591022005745
Line 83 to 86: Please add the missing reference:
https://www.sciencedirect.com/science/article/pii/S0032591022003473
Line 123: Please indicate which software or code did you use or if it is an in-house code.
Line 231: Please add the definitions of LUP, LMID and LDWN.
Line 259: The top size of the PSD should be controlled by the OSS, there are particles that pass thought the crushing chamber by the open side.
Table 1: Please add the reference or how were calculated the contact and breakage parameters of the model
Figure 7: What is the unit of measurement for wear intensity?
Round 2
Reviewer 4 Report
Comments and Suggestions for Authors
Major
- The rolling friction coefficient is not listed, as it is an important parameter when modeling spherical particles.
- There is no clear relationship between the breakage parameters of the reference [18] and the material used in the crusher. Bruchmüller et al. did not use porphyry copper deposits for the parameter calibration of their breakage model.
- Please elaborate the idea that your model can predict wear but not power. Both variables are calculated with the contact forces between the particle and the mantle; therefore, if the power cannot be predicted, the wear is not representative.
Minor
- Eq. 3: Please fix the use of bold variables in the equations.
